# Natural Products in Neurodegenerative Diseases: A Great Promise but an Ethical Challenge

**DOI:** 10.3390/ijms20205170

**Published:** 2019-10-18

**Authors:** Marco Di Paolo, Luigi Papi, Federica Gori, Emanuela Turillazzi

**Affiliations:** Section of Legal Medicine, Department of Surgical Pathology, Medical, Molecular and Critical Area, University of Pisa, 56124 Pisa, Italy; marco.dipaolo@unipi.it (M.D.P.); luigi.papi@unipi.it (L.P.); gori.federica@virgilio.it (F.G.)

**Keywords:** neurodegenerative diseases, natural products, ethics, patients’ autonomy, beneficence, nonmaleficence, medical liability

## Abstract

Neurodegenerative diseases (NDs) represent one of the most important public health problems and concerns, as they are a growing cause of mortality and morbidity worldwide, particularly in the elderly. Despite remarkable breakthroughs in our understanding of NDs, there has been little success in developing effective therapies. The use of natural products may offer great potential opportunities in the prevention and therapy of NDs; however, many clinical concerns have arisen regarding their use, mainly focusing on the lack of scientific support or evidence for their efficacy and patient safety. These clinical uncertainties raise critical questions from a bioethical and legal point of view, as considerations relating to patient decisional autonomy, patient safety, and beneficial or non-beneficial care may need to be addressed. This paper does not intend to advocate for or against the use of natural products, but to analyze the ethical framework of their use, with particular attention paid to the principles of biomedical ethics. In conclusion, the notable message that emerges is that natural products may represent a great promise for the treatment of many NDs, even if many unknown issues regarding the efficacy and safety of many natural products still remain.

## 1. Introduction 

Neurodegenerative diseases (NDs) include a number of chronic progressive disorders of the central nervous system that are caused by the degradation and subsequent loss of neurons. NDs represent one of the most important public health problems and concerns, as they are a growing cause of mortality and morbidity worldwide, particularly in the elderly. The aging of the population has contributed to the increase of NDs [1,2], and age-related diseases such as NDs are becoming extremely important, due to their irreversibility, lack of effective treatment, and accompanied social and economic burdens [3].

Traditionally, classifications of NDs included Parkinson’s disease, which is well characterized by a loss of dopaminergic nigrostriatal neurons; Huntington’s disease, in which the loss of spiny, medium-sized striatal neurons occurs; and Alzheimer’s disease (AD), due to diffuse cerebral atrophy. Other disorders such as primary dystonia or essential tremor were also referred to as NDs [4]. NDs recognize a broad, often overlapping, spectrum of symptoms, varying from memory and cognitive deficits to the impairment of a person’s ability to move, speak, and breathe; they also share some clinical characteristics such as a relentless progression over years, sometimes even decades [5].

Beyond the known differences in the pathogenic mechanisms of individual diseases, neurodegeneration, understood as the chain of events leading to gradual loss of neurons’ functional properties until cell death, represents the key point of this group of diseases [4], and attracts research efforts in trying to understand precise pathogenic mechanisms and achieve valid therapies.

In fact, despite remarkable breakthroughs in our understanding of NDs, there has been little success in developing effective therapies [6]. The therapies currently available seem to be inadequate for NDs, as they only act to alleviate symptoms but cannot stop the progress of the disease. The use of natural products (NPs) is growing, probably due to several factors [7,8] (Figure 1).

They may offer great potential opportunities in the prevention and therapy of NDs [9], and scientists are increasingly exploring options with herbal drugs and natural products [10]. As for many traditional drugs, many clinical concerns have arisen regarding the use of NPs, mainly focusing on the lack of scientific support or evidence for their efficacy and patient safety. These clinical uncertainties raise critical questions from a bioethical and legal point of view, as considerations relating to patient decisional autonomy, patient safety, and beneficial or non-beneficial care may need to be addressed, meaning that many intriguing points may arise regarding the use of NPs [11,12].

This paper does not intend to advocate for or against the use of natural products but to analyze the ethical framework of their use, with particular attention paid to the principles of biomedical ethics as described by Beauchamp and Childress [13], then addressing the strict intertwining of bioethics, safety, and responsibility related to the use of natural products.

## 2. Natural Products in Neurodegeneration

Despite specific clinical and etiopathogenic differences, NDs show some common features such as abnormal protein deposition, abnormal cellular transport, mitochondrial deficits, inflammation, intracellular Ca2+overload, uncontrolled generation of ROS, and excitotoxicity, thus suggesting the existence of converging pathways of neurodegeneration and reinforcing the importance of these pathways as common targets for intervention strategies [14]. Furthermore, reactive astroglia and/or microglia have been implicated in the pathogenesis of all major neurodegenerative disorders [15].

Over the years, target-based therapies such as neurotransmitter modulators, direct receptor agonists/antagonists, second messenger modulators, stem cell-based therapies, hormone replacement therapy, and neurotrophic factors, as well as regulators of mRNA synthesis and their translation into disease-causing mutant proteins, have been introduced and implemented [14].

However, there are currently no therapeutic strategies capable of either preventing or reducing the progression of NDs; many of the approved drug regimens for NDs help to treat the symptoms but do not cure the disease itself. As many of the traditional symptomatic therapies may lose their effectiveness over time, produce disruptive symptoms of their own, and show severe side effects [16], there is an urgent need to develop more effective and safer therapies that can be employed over a long duration of NDs.

Several natural agents have been proposed to complete and/or assist the traditional pharmacological agents in the treatment of neurodegenerative disorders, and the general idea of this is provided, among others, by Srivastava and Yadav [17]. Their use in NDs is widely reported in the literature [18,19,20,21], as these products show several different neuroprotective activities. Mitochondrial dysfunction, apoptosis, excitotoxicity, inflammation, oxidative stress, and protein misfolding are among the main neuroprotective targets of natural products [21,22,23,24,25].

Animal-based products such as omega-3 fatty acids inhibit cellular toxicity and show anti-inflammatory effects in AD [26]. Plant derived products like Lunasin, Polyphenols, Alkaloids, and Tannins are potential therapeutic candidates for AD [25]. Resveratrol and flavonoids seem to be dietary components with specific neuroprotective action and positive effects on human cognitive decline [27,28].

It is beyond the scope of this paper to analyze the pharmacological and pharmacodynamic profiles of the various natural products proposed for the treatment of NDs. However, some general concepts are necessary because, in addition to having important clinical repercussions, they reflect heavily on the bioethical—and even possibly legal—implications of the administration of natural products in NDs.

It can be said that natural products may be very promising due to their anti-neurodegenerative action, with the potential to treat a large number of patients worldwide.

A general belief that NPs are safe exists: Many patients take NPs, presumably based on the assumption that they are effective, safe, and less toxic than traditional drugs [29]. Very often, patients assess NPs as safer than biomedicine on the basis of being “natural”. A factor promoting the consumption of herbal products may be the preconception that botanicals are natural and “natural is good” and consequently safe.

However, as drugs, they either may have adverse effects or may be not effective [30]. Because natural products may derive from diverse biological sources, their conversion into therapies is not trivial. Challenges may include concerns regarding their stability and neuroavailability [22], difficulty in adequately identifying and quantifying the active principle, and, finally, the difficulty of organizing large clinical trials to test these complex products.

Product characterization may be a key problem for NPs, as for many of them, the specific bioactive components have not been identified or are not fully characterized [21]. Herbal products contain complex active components or phytochemicals such as flavonoids, alkaloids, and isoprenoids. Therefore, it is frequently difficult to determine which component of the herb has the most biological activity [31]. A further complication is the fact that several natural compounds have limited stability and are easily degradable and may be metabolized to inactive products [22,32,33]. Concerns regarding compound solubility, restricted passage through the blood–brain barrier, and availability exist [22,33,34]. Furthermore, the evidence for the potential protective effects of selected herbs is generally based on experiments demonstrating a biological activity in a relevant in vitro bioassay or experiments using animal models [35]. However, in order to further widen their acceptance and use, clinical trials should be encouraged [36,37,38], since for many products, a translation to clinical trials may offer challenges that need to be addressed [39].

Firstly, in clinical trials, single compounds are more frequently investigated, while the investigation of plant extracts containing a variety of secondary metabolites is more common in studies prior to clinical studies. The combination of the various active principles in extracts can lead to additional or synergistic effects, giving better antioxidant/disease-modifying activity [39]. Moreover, natural antioxidants (i.e., from natural products or plant extracts) could also share this multi-target drug profile, and the combination of single compounds or extracts also needs to be further investigated.

Concerns regarding the purity and potency of herbal products exist. Product quality is influenced by many factors, including which portion of the plant is used (i.e., the root, stem, leaves, flowers), the time of harvest (i.e., young versus old plants), the handling of the product, and the proper identification of the plant. Furthermore, labelling may be inaccurate [39]. Of course, the concerns regarding the quality of herbal products cannot be generalized, and it is beyond question that high-quality products also exist in the legal market and that many manufacturers are beginning to significantly improve their quality standards. Nevertheless, the quality of herbal products still varies from one product to another, and many companies still sell low-quality products [40]. The focus on the quality of the products on the market is closely linked to the regulatory framework for such preparations [41], since a product marketed without any medical, pharmaceutical, or other regulation is bound to be very different from one which is regulated, for example, in the context of a licensing or registration scheme.

Conclusively, natural products have recently gained greater attention as alternative or integrative therapeutic agents against AD and other NDs [42]. However, critics have raised concerns that the popularity of natural products is growing without scientific support or evidence of their efficacy and safety [11].

These uncertainties justify both paying a great deal of attention to these products, and the careful monitoring of their use by clinicians. At the same time, the use of NPs often requires clinicians to make decisions under conditions of uncertainty, thus involving questions about which ethical principles clinical decision-making should rely on, what kind of information should be provided to patients, and what obligations arise on the part of physicians (Figure 2).

## 3. Natural Products: On the Cutting Edge of Ethics?

### 3.1. Patient Autonomy

Autonomy is a fundamental bioethical principle requiring that a person has the capacity and opportunity to act autonomously; that is, to freely and voluntarily make choices. In a health-care setting, when a patient exercises her/his autonomy, she/he decides which of the options for dealing with her/his health-care problems will be best, given her/his values, concerns, and goals. A patient who makes autonomous choices is able to opt for what she/he considers will be best, all things considered [43].

It is increasingly clear that several gaps may exist regarding the use of natural products that may lead to a failure to provide adequate information regarding natural products and leave patients in the dark. The goal of the informed consent process is to provide sufficient information to a patient so that she/he can make the voluntary decision whether or not to take a natural product as an alternative or in combination with other types of drug treatment. Obtaining consent involves informing the subject about the potential risks and/or benefits of the proposed product and the alternative treatments available, if any. A description of any foreseeable risks or discomforts to the subject, an estimate of their likelihood, a description of any benefits that may reasonably be expected from the natural product, and, finally, the disclosure of any appropriate alternative procedures or courses of treatment that might be advantageous to the patient, should be the milestones of the informative process.

Unfortunately, several disturbing points about the use of natural products can compromise patient autonomy.

First of all, there is a lack or scarcity of basic information regarding the safety and effectiveness of many natural products used in NDs. In their analysis of the risks of complementary and alternative medicine, White et al. [44] clearly highlighted that natural products, such as plant extracts, may be pharmacologically complex, and so have multiple physiological effects which may represent a beneficial synergy or harmful interaction, depending on the specific context. Natural products may have pharmacological effects, just as with synthetic pharmaceuticals [45]; however, they are generally perceived as safe due to being natural, and patients themselves are less likely to report harmful incidents that may be associated with natural products than with conventional drugs [46]. Furthermore, in many cases of natural products, the associated risks are really unknown, and there is a limitation of currently available information concerning their safety and efficacy [47].

The informative process regarding the use of natural products, both as primary treatment and as adjunct treatment, may be complex and multifaceted as the evidence regarding natural products safety and efficacy is still not consolidated, even if it is growing, and many disturbing points (insufficient detailed knowledge of natural products pharmacology and drug interactions) may represent great challenges in obtaining real, informed consent regarding the use of natural products themselves, since they are not embedded in well-understood scientific paradigms. Taking these concerns to extremes, some authors even went as far as to state that it is unethical for medical professionals to offer or endorse “alternative medicine” treatments for which there is no known causal mechanism [48].

As the ethical concept of treating patients as people with respect and with the understanding of the individual’s right to self-determination (autonomy) is fundamental [49,50], removing any ambiguity from the informative process is, at the moment, the only ethically correct answer. When uncertainties regarding efficacy and safety exist, as for many natural products, honest information should include telling patients about the degree of uncertainty associated with the efficacy and safety of the treatment, as well as the availability and risk–benefit ratio of other treatment options [51], so that patients could become comfortable with uncertainties and accept or refuse treatment with natural products. As such, the use of natural products in NDs could even result in a net gain in patient autonomy, as it provides patients with different therapeutic options with respect to traditional drug therapies. In other words, despite the existing gaps of knowledge on the efficacy and safety of NPs, patients affected by NDs should be able to freely choose to undergo treatments with NPs, with the hope of improving their quality of life or lengthening survival.

The final thought is that also in treatments through natural products does the culture of respect of patient autonomy, preference, and choice provide the underpinning needed to establish an effective physician–patient relationship, in which there may also be space for a conscious adherence to therapies which, although not yet rigorously validated, may nevertheless represent for the patient a beneficial alternative or supplement to traditional treatments—unfortunately, of poor effectiveness.

This paves the way for further reflection concerning the accurate assessment of patients’ cognitive ability to give informed consent. A proper informative process requires that the patients “understand” the information given by the physician and appreciate the consequences of their decision. In patients affected by NDs, it may be difficult to evaluate whether the patient is able to understand the information presented and consent to or refuse the proposed treatment [52]; in other words, the assessment of the decision-making capacity, defined as the ability to understand and reason through the decision-making process, may be challenging in ND patients. This may represent a problem of great magnitude if we only consider that the numbers of older adults with cognitive impairment due to NDs is estimated to be high and rapidly increasing [53]. Case by case, the physician has the legal and ethical duty to explore the real existence of incompetence and if—and to what extent—it may affect the capacity for decision making for consent to treatment. Legal standards for decision-making capacity may be different across national jurisdictions; however, they generally include the capacity to understand the relevant information regarding a proposed treatment, its consequences, and alternatives [54,55]. A personalized, patient-targeted approach, grounded on the individual patient’s characteristics and clinical situation, should guide physicians while assessing the patient’s level of decisional capacity [53,54,55]. If the physician believes that a patient is incompetent to make a treatment decision, advance directives or legal proxies must be considered according to the existing national regulatory framework.

### 3.2. Beneficence/Nonmaleficence/Justice

The ethical concepts of beneficence, nonmaleficence, and justice warrant a shared discussion concerning the use of natural products, as they appear to be strictly interconnected.

Beneficence and the twin concept of nonmaleficence demand that patients should not be harmed by a treatment, both entreating the physician to avoid the causation of harm, and to provide benefits as well as balance benefits, burdens, and risks. The justification for the use of natural products seems to be that providing ND patients with these products provides them with benefits and could improve their quality of life, and even prolong their lives.

The central ethical challenge in the use of NPs is thus to determine when evidence has reached a sufficient level of certainty to warrant clinical introduction. In considering the available evidence, relevant factors include the scope of estimated benefit, the existence of alternative treatments, the nature and scope of potential harms, and the overall quality of evidence. When only limited evidence supports the use of natural products, the appropriate support of clinician (and patient) choice remains an ethical concern. Rarely does a single response exist. The clinically (and, of consequence, ethically) justified use of any natural product must be individualized to the patient’s circumstances, including the stage of the disease, the severity of symptoms affecting the quality of life, and the existence/absence of valid therapeutic alternatives. Given the evidence gaps that still exist for many NPs, clinicians should consider what uncertainties they (and patients) are most comfortable with. 

In the use of NPs, the physician’s ethical obligation of beneficence/nonmaleficence (an obligation to maximize benefit and minimize harm) does not differ from standard clinical practice, where every drug is always potentially risky. When the risk of harm is disproportional to the potential benefit, providing the patient with this product may be questionable in the light of the ethical principles of beneficence/nonmaleficence.

Justice demands that all patients be treated fairly since “justice [is] fair, equitable, and appropriate treatment in light of what is due or owed to persons. Injustice involves a wrongful act or omission that denies people resources or protections to which they have a right” [13].

As mentioned above, this may be an option for ND patients who set their hopes on NPs as an alternative or supplement to the traditional synthetic drugs; however, this principle is only valid if and when patients do indeed get access to NPs.

Data from the literature recall the fact that the use of natural products within complementary and alternative medicine has been shown to be a popular choice of therapy among patients [29,56,57,58,59]. It has been reported that 80% of the world’s population uses natural products, rising to 95% of the population in developing countries [56]. The use of herbal medicine was commonly reported across the European Union according to the CAMbrella consortium [57]. Patients with chronic diseases that are mostly resistant to conventional therapies tend to choose alternative and integrative therapies [58].

However, differences in NP utilization emerge for reasons that have to be discussed, as they seem to be unacceptable from an ethical point of view, thus representing red flags regarding the justice principle. Differences in utilization could be either for acceptable reasons (e.g., personal preferences and choices) or unacceptable reasons, such as costs and opportunities.

As Nissen et al. outlined [56], herbal and natural products may be the only available treatment for low-income people in developing countries; on the other hand, in high-income countries, natural products are often provided outside public healthcare services or insurance coverage, thus being mostly used by educated citizens of working age and with an above-average income. It has been reported that in industrialized societies, the use of complementary medicine has been found to be associated with higher income and higher education, and people who have lower incomes and educational levels tend not to use complementary medicine [60]. As with many other services of alternative and complementary medicine, in most countries worldwide, NPs seem to be mostly provided outside the public health system, and patients have to pay themselves for these products [56,58].

Immediate questions of ethics arise: If NPs in NDs represent a clinically prospective option, why do people who have free traditional health-care services or insurance coverage for traditional treatments have to pay out-of-pocket for NPs? Additionally, in those countries where inadequate and expensive conventional medical services exist, conventional medical care may not be accessible to the poorest minorities. In these situations, NPs are not alternative nor integrative [60]: They may represent the only therapeutic option, thus facilitating a “separate but unequal care system” [61]. In other words, the above dimensions of the use of NPs point out issues of inequity in gaining access to NPs, as whether patients affected by NDs who set their hopes on NPs are able to access them may be determined by economic and financial factors outside the patients’ control, making this access unfair.

A further argument that may be subject of ethical discussion is that public funding should not be allocated to research of implausible treatments, representing a potential waste of resources both human and economic, and an additional expense over and above other healthcare costs. One of the key themes is that, given the scarcity of resources that can be allocated to research, every effort should go to those areas where reasonable, good evidence also exists [60].

However, this may represent a double-edged sword, ethically speaking: The further development of NPs is achievable only on a broad base of quality research. For example, the National Centre for Complementary and Alternative Medicine experience in the United States has shown that when funds are available and priorities are set, research on alternative and complementary medicine will grow exponentially [60,61]. The status of the research on complementary and alternative medicine in the European Union is not encouraging: Complementary and Alternative Medicine research in Europe is not well-funded by the countries or research organizations, and is in large part charitably supported [57].

This seems to feed a vicious circle, since the conduct of high-quality research on complementary and alternative medicine requires a commitment by the research community, as well as sustained financial support from governments and industry [62]. Implementing research on the use of NPs, as well as other alternative and complementary medicine, is pivotal to the achievement of “high-quality” evidence in support of the use of NPs.

## 4. Medical Liability Scenarios

As the use of alternative therapies and natural products grows, there is likely to be heightened concern about the liability implications of delivering these therapies. It has been reported that complementary and integrative medicine (in which the use of NPs is included) have not, until now, been the subject of serious malpractice litigation [63]; however, herbal medicines and other supplements are one of the more controversial fields of medicine [64].

Generally speaking, medical malpractice occurs if a physician fails to meet the standard of care established and negligently injures the patient, requiring three elements: (1) The patient suffered damage; (2) the negative event proceeded directly by the action or inaction of the healthcare provider; and (3) the physician was negligent, which essentially entails showing that she/he took less care than that which is customarily practiced by the average member of the profession in good standing, given the circumstances of the doctor and the patient [65].

These elements are long-established fundamental principles; however, the manner in which these principles will be applied to the use of NPs can raise important questions.

First of all, what is the standard of care for these treatments? Has the physician met the applicable standard of care while treating the patient with NPs? This may represent a difficult conceptual passage, as the issue of standard of care (i.e., the degree of care, diligence, and skill which is provided by a reasonable physician under like or similar circumstances) is not clearly defined for complementary and alternative treatments [64,66]. In some activities included in alternative or complementary practices, it may be difficult to establish the standard appropriate in the discipline, as a well-established scientific base and standardized protocols are generally lacking.

However, we can speculate that a physician who prescribes NPs would be held to a common standard in medical practice when assessing what she/he knew or should have known regarding the safety and the efficacy of the prescribed remedies [64,66]. Prescriptions of natural products, although not negligent per se, could represent a deviation from the standard of care, if it is demonstrated that this prescription was not something a reasonable physician would have done in similar circumstances and with the same patient.

The conceptual framework of potential alleged malpractice claims regarding the use of natural products focuses on some key points: (i) Whether the evidence in the scientific literature supports, does not support, or is inconclusive regarding the use of a natural product along the dimensions of safety and efficacy; and (ii) an accurate evaluation of the risk–benefit balance, including the stage and severity of the illness, the curability of the illness with conventional therapy, its potential toxicity, and adverse effects [63,64,67,68].

Although a standard of care for the use of NPs has not yet been clearly defined, as many therapies based on natural products have not yet been the subject of rigorous, controlled studies, the care provided through these products could be judged by either the clinician’s duty to carefully consider findings from conventional medicine, the clinical status of that patient, the existing evidence on the natural products proposed, the reported adverse events, and, finally, the duty of carefully monitoring the clinical evolution of the patient.

As a concluding statement, it would seem appropriate to reiterate the words of Cohen, who suggested that “If evidence supports safety, but evidence regarding efficacy is inconclusive—accept but monitor; if the medical evidence supports efficacy, but evidence regarding safety is inconclusive—accept but monitor; and the medical evidence indicates either serious risk or inefficacy—avoid and discourage” [69].

## 5. Concluding Remarks

In conclusion, the notable message that emerges is that NPs may represent a great promise for the treatment of many NDs, where traditional therapies via synthetic drugs only act to alleviate symptoms, but cannot stop the progress of the disease, and thus are substantially inadequate.

It is, however, problematic for this message that there are still many unknown issues regarding the efficacy and safety of many natural products. There is much yet to be investigated, characterized, and learned. This message strongly underscores urgent needs. Clinicians must routinely inquire about all product use—conventional, complementary, and alternative—to promote patient safety and ethical care.

The evaluation of patient safety and of the efficacy of natural products should represent the guiding principle of physicians’ conduct.

Finally, researchers should actively start to expand the knowledge base regarding natural product safety and efficacy, emphasizing that fundamental research to advance the understanding of the basic biological mechanisms of action of these products is pivotal.

## Figures and Tables

**Figure 1 ijms-20-05170-f001:**
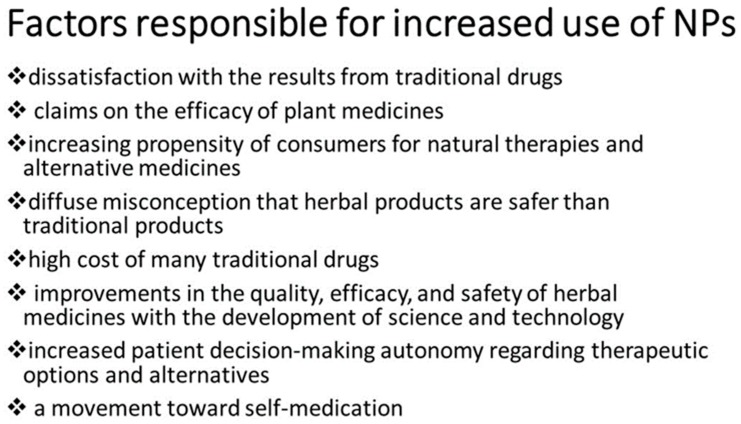
Factors responsible for increased use of NPs.

**Figure 2 ijms-20-05170-f002:**
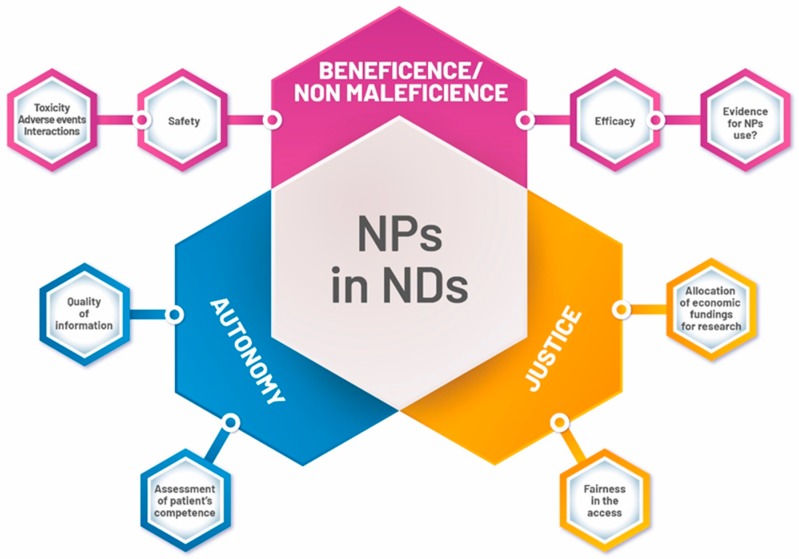
Main ethical topics of Natural Products (NPs) use in NDs.

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
