# Peer review of "Natural Products in Neurodegenerative Diseases: A Great Promise but an Ethical Challenge"

_ijms, 2019, doi:10.3390/ijms20205170_

Round 1

Reviewer 1 Report

Natural products in neurodegenerative diseases: a great promise but an ethical challenge

Marco Di Paolo, Luigi Papi, Federica Gori and Emanuela Turillazzi

Herewith I am submitting my reviewer comments for the above-mentioned manuscript, which is under consideration to be published in International Journal of Molecular Sciences. The article is a review paper about natural products in neurodegenerative diseases. In many places the article is vague or non-scientific. Some statements are jus wrong. In addition to that, the article misses the point of a review paper to my opinion. A review should give the reader an easier access and overview over the existing literature. To my opinion this article doesn’t do that. It isn’t well structured or organized so to me some key parameters that make a good review are just not met.

The article contains only one figure and no tables. Figures and tables are a great tool to help digest the topic for the readers. So I would highly recommend making use of this.

Line 27: The sentence starting with “Neurodegenerative diseases (NDs) include a number” is too long and thus difficult to read.

Line 32: “age – related disease like NDs are becoming” should be “age – related diseases like NDs are becoming”

Line 50: “However, many clinical concerns have arisen regarding the use of NPs, mainly focusing on lack of scientific support or evidence for efficacy and patients’ safety.” Shouldn’t we, if we consider natural products (whatever that means) seriously as drugs treat them in the same way as we would treat any other drugs? Or in other words do we really need a special treatment for these?

Line 77: The sentence starting with “Their use in NDs is widely reported in literature” is too long and thus difficult to read.

Line 86: “For example, natural products like Lunasin, Polyphenols, Flavonoids, Alkaloids, and Tannins are potential therapeutic candidates for Alzheimer disease (AD) [21]” how do you define natural products here? These are all chemicals that can be synthesized or extracted from plants. So is it natural only if you give the entire plant? Something that comes from a plant? I think you could be more precise here.

Page 98: “They are generally considered to be less toxic, with less side‐effects and less expensive than synthetic drugs.” That is how people perceive it often. But it is not true at all. Just if something is “natural” (whatever that means) doesn’t make it any more or less toxic at all. Some of the most toxic substances there are (like botulin’s toxin or snake venoms for instance). The notion that natural means less toxic comes from being scientifically ill informed. I don’t want to see any encouragement of wrong stereotypes in an actual scientific paper.

Line 100: “However, because natural products may derive from diverse biological sources, their conversion into therapies is not trivial.” Including synthetic materials into therapies is not exactly trivial either.

Line 111: “Finally, as recently reviewed by Pohl and Kong Thoo Lin [29], the translation of the satisfactory results on in vivo (animal models) and in vitro models to human patients is still inconclusive and demonstrates limited success.” A lot of things that work in-vivo on animals just doesn’t work in humans (regardless of what the drug actually is)

Line 126: “as they are thought to be less toxic and more effective than novel synthetic drugs” NO, JUST NO.

Fig 1: I don’t really understand the logic behind the figure. What do the different connections mean? Also the organization is chaotic. Line thicknesses, size and shapes of circles, distances between circles all differ without an apparent meaning to it?

Line 172: “Taking these concerns to extremes, some authors even went so far as to state that it is unethical for medical professionals to offer or endorse ‘alternative medicine’ treatments, for which there is no known causal mechanism [37].” There is no such thing as “alternative medicine”. If it works it is just medicine.

Line 219: “The central ethical challenge in the use of NPs is thus to determine when evidence has reached a sufficient level of certainty to warrant clinical introduction.” This is established and there are guidelines for these things for any drugs. I do not see the point of making this different from any other drug approval process?

Line 262: “Immediate questions of ethics arise: if NPs in NDs represent a clinically prospective option, why people who have free traditional health care services or insurance coverage for traditional treatments have to pay out – of pocket for NPs?” very often insurances do not cover certain products because they have proven not to work! To my opinion this is actually a good thing. It is our responsibility as scientists to inform the public about what works and what doesn’t (or for which things we don’t know)

Line 332: “In conclusion, the notable message that emerges is that NPs may represent a great promise for the treatment of many NDs, whose traditional therapies via synthetic drugs only act to alleviate symptoms but cannot stop the disease progress, thus being substantially inadequate.” Should this imply that natural substances are the solution to this problem? Why?

Author Response

We thank the reviewer 1 for the criticims and suggestions, aimed to improve the paper.

Point 1.

Line 27: The sentence starting with “Neurodegenerative diseases (NDs) include a number” is too long and thus difficult to read.

We shortened the sentence.

Point 2.

Line 32: “age – related disease like NDs are becoming” should be “age – related diseases like NDs are becoming”.

We modified accordinlgly and changed disease into diseases.

Point 3.

Line 50: “However, many clinical concerns have arisen regarding the use of NPs, mainly focusing on lack of scientific support or evidence for efficacy and patients’ safety.” Shouldn’t we, if we consider natural products (whatever that means) seriously as drugs treat them in the same way as we would treat any other drugs? Or in other words do we really need a special treatment for these?

We agree with the reviewer regarding the fact that natural products, being pharmacologically active, need to be treated similarly to conventional medicines. This is a need in the context of the general and clinical use of medicinal plants and products derived from them. We changed the sentence. And added “as for many traditional drugs”. See line 57 of the revised version.

Point 4.

Line 77: The sentence starting with “Their use in NDs is widely reported in literature” is too long and thus difficult to read.

We changed the text accordingly, eliminating a redundant part. See lines 86-89 of the revised version.

Point 5.

Line 86: “For example, natural products like Lunasin, Polyphenols, Flavonoids, Alkaloids, and Tannins are potential therapeutic candidates for Alzheimer disease (AD) [21]” how do you define natural products here? These are all chemicals that can be synthesized or extracted from plants. So is it natural only if you give the entire plant? Something that comes from a plant? I think you could be more precise here.

We agree with the reviewer regarding the potential misinterpretation of the term “natural products” and changed the sentence. See lines 91-94.

Point 6.

Page 98: “They are generally considered to be less toxic, with less side‐effects and less expensive than synthetic drugs.” That is how people perceive it often. But it is not true at all. Just if something is “natural” (whatever that means) doesn’t make it any more or less toxic at all. Some of the most toxic substances there are (like botulin’s toxin or snake venoms for instance). The notion that natural means less toxic comes from being scientifically ill informed. I don’t want to see any encouragement of wrong stereotypes in an actual scientific paper.

We apologize if we gave the impression of having said this. It may have been due to a bad translation of our thinking. The paper is aimed at highlighting the issue of safety and efficacy of the NPs. However we have included some explanatory and in-depth sentences. See lines 102-106 of the revised version.

Point 7/8

Line 100: “However, because natural products may derive from diverse biological sources, their conversion into therapies is not trivial.” Including synthetic materials into therapies is not exactly trivial either.

Line 111: “Finally, as recently reviewed by Pohl and Kong Thoo Lin [29], the translation of the satisfactory results on in vivo (animal models) and in vitro models to human patients is still inconclusive and demonstrates limited success.” A lot of things that work in-vivo on animals just doesn’t work in humans (regardless of what the drug actually is).

Once again, we agree with the reviewer that also for synthetic materials the clinical translation is complex. Being a paper dedicated to the NPs, we limited ourselves to underline that this problem exists also for these products. We added some sentences in this part of the paper. See lines 119-123 of the revised version.

Point 9.

Line 126: “as they are thought to be less toxic and more effective than novel synthetic drugs” NO, JUST NO.

Once again, we apologize if we gave the impression of having said this. It may have been due to a bad translation of our thinking. All the paper is aimed at highlighting the issue of safety and efficacy of the NPs. We believe that the sentence following the one disputed by the reviewer should be a clear demonstration of this. However, we deleted the sentence.

Point 10.

Fig 1: I don’t really understand the logic behind the figure. What do the different connections mean? Also the organization is chaotic. Line thicknesses, size and shapes of circles, distances between circles all differ without an apparent meaning to it?

We modified the figure.

Point 11.

Line 172: “Taking these concerns to extremes, some authors even went so far as to state that it is unethical for medical professionals to offer or endorse ‘alternative medicine’ treatments, for which there is no known causal mechanism [37].” There is no such thing as “alternative medicine”. If it works it is just medicine.

The point underlined by the reviewer is very thorny; indeed many uncertainties still exist on the exact definition of CAM and the general idea may be that registered/licensed herbalmedicinal products are not part of Complementary and Alternative Medicine. They are pharmacologically active medicines and need to be treated similarly to conventional medicines. Our refrence to the paper by Shahvisi was only aimed to highlight the difficulties of the information process when using substances whose efficacy and safety there is still no undisputed evidence. It is true this also happens for many traditional drugs.

Point 12.

Line 219: “The central ethical challenge in the use of NPs is thus to determine when evidence has reached a sufficient level of certainty to warrant clinical introduction.” This is established and there are guidelines for these things for any drugs. I do not see the point of making this different from any other drug approval process?

We agree with the reviewer: any difference with traditional drugs approval should exist. The question is that many natural products are still not regulated in their marketing process.

Point 13.

Line 262: “Immediate questions of ethics arise: if NPs in NDs represent a clinically prospective option, why people who have free traditional health care services or insurance coverage for traditional treatments have to pay out – of pocket for NPs?” very often insurances do not cover certain products because they have proven not to work! To my opinion this is actually a good thing. It is our responsibility as scientists to inform the public about what works and what doesn’t (or for which things we don’t know).

It may be a good thing; however the ethical concern relating to justice principle remains and the existing literature on this issue demonstrates it.

Point 14.

Line 332: “In conclusion, the notable message that emerges is that NPs may represent a great promise for the treatment of many NDs, whose traditional therapies via synthetic drugs only act to alleviate symptoms but cannot stop the disease progress, thus being substantially inadequate.” Should this imply that natural substances are the solution to this problem? Why?

We have never claimed that NPs could represent the solution to the problem of treating neurodegerative diseases. We would not be able to say this since we are neither clinicians nor pharmacologists. We have only acknowledged from the abundant literature that they can represent a hope and a promise for certain patients. And deliberately we never used the word solution in our text.

Finally, we added a figure (Fig.1) and English editing process was arranged by MDPI.

Reviewer 2 Report

Dear Editor,

I carefully read the critical review by Di Paolo and co-workers, which is interesting and overall balanced in its contents. Some comments in order to improve the current manuscript:

English language needs to be improved (i.e. actually there are some typos in the paper) Lines 111-113: "The
112 translation of the satisfactory results on in vivo (animal models) and in vitro models to human
113 patients is still inconclusive and demonstrates limited success". Therefore, more clinical trials are needed. Please, the Authors should be more precise and better discuss the present statement. Lines 121-124: Authors refer to concerns about the quality of the marketed nutraceutical products. However, this consideration is not accurate at all. Actually, there are many high quality products available. The critical point is that the nutraceuticals' marketing should be more regularized by legislation and only products deemed safe should be placed on the market. Authors should consider in their review to refer to doi: 10.1016/j.phrs.2017.12.029, doi: 10.14283/jpad.2016.10, doi: 10.5114/aoms.2019.85463 and  doi: 10.2174/1381612824666171213100449.

Author Response

We thank the reviewer 2 for very stimulating comments. We modified the paper accordingly.

Point. 1

English language needs to be improved (i.e. actually there are some typos in the paper).

Language editing was arranged by MDPI.

Point 2

Lines 111-113: "The translation of the satisfactory results on in vivo (animal models) and in vitro models to human patients is still inconclusive and demonstrates limited success". Therefore, more clinical trials are needed. Please, the Authors should be more precise and better discuss the present statement.

We changed the text into “Furthermore, the evidence for the potential protective effects of selected herbs is generally based on experiments demonstrating a biological activity in a relevant in vitro bioassay or experiments using animal models [35]. However, in order to further widen their acceptance and use, clinical trials should be encouraged [36-38], since for many products, a translation to clinical trials may offer challenges that need to be addressed [39]. See lines 119-123 of the revised version. We added the relevant references.

Point 3

Lines 121-124: Authors refer to concerns about the quality of the marketed nutraceutical products. However, this consideration is not accurate at all. Actually, there are many high quality products available. The critical point is that the nutraceuticals' marketing should be more regularized by legislation and only products deemed safe should be placed on the market.

We changed the text into “Of course, the concerns regarding the quality of herbal products cannot be generalized, and it is beyond question that high-quality products also exist in the legal market and that many manufacturers are beginning to significantly improve their quality standards. Nevertheless, the quality of herbal products still varies from one product to another, and many companies still sell low-quality products [40]. The focus on the quality of the products on the market is closely linked to the regulatory framework for such preparations [41], since a product marketed without any medical, pharmaceutical or other regulation is bound to be very different from one which is regulated, for example, in the context of a licensing or registration scheme.” See lines 134-142 of the revised version. We added the relevant references.

Point 4

Authors should consider in their review to refer to doi: 10.1016/j.phrs.2017.12.029, doi: 10.14283/jpad.2016.10, doi: 10.5114/aoms.2019.85463 and doi: 10.2174/1381612824666171213100449.

We thank the review for having expanded our knowledge on the subject and we added the suggested references. See references 27, 28, 37,38.

Round 2

Reviewer 1 Report

The authors have done a good job to improve the article. I am still not very amazed by it but it is better written now and the claims that I found somewhat unscientific or misleading are much better now.